# Natural Killer Cell-Derived Extracellular Vesicles as a Promising Immunotherapeutic Strategy for Cancer: A Systematic Review

**DOI:** 10.3390/ijms24044026

**Published:** 2023-02-16

**Authors:** Alvin Man Lung Chan, Jin Min Cheah, Yogeswaran Lokanathan, Min Hwei Ng, Jia Xian Law

**Affiliations:** 1Centre for Tissue Engineering and Regenerative Medicine, Faculty of Medicine, Universiti Kebangsaan Malaysia, Kuala Lumpur 56000, Malaysia; 2Ming Medical Sdn Bhd, D3-3 (2nd Floor), Block D3 Dana 1 Commercial Centre, Jalan PJU 1a/22, Petaling Jaya 47101, Malaysia

**Keywords:** natural killer cells, extracellular vesicles, exosomes, immunotherapy, cancer

## Abstract

Cancer is the second leading contributor to global deaths caused by non-communicable diseases. The cancer cells are known to interact with the surrounding non-cancerous cells, including the immune cells and stromal cells, within the tumor microenvironment (TME) to modulate the tumor progression, metastasis and resistance. Currently, chemotherapy and radiotherapy are the standard treatments for cancers. However, these treatments cause a significant number of side effects, as they damage both the cancer cells and the actively dividing normal cells indiscriminately. Hence, a new generation of immunotherapy using natural killer (NK) cells, cytotoxic CD8^+^ T-lymphocytes or macrophages was developed to achieve tumor-specific targeting and circumvent the adverse effects. However, the progression of cell-based immunotherapy is hindered by the combined action of TME and TD-EVs, which render the cancer cells less immunogenic. Recently, there has been an increase in interest in using immune cell derivatives to treat cancers. One of the highly potential immune cell derivatives is the NK cell-derived EVs (NK-EVs). As an acellular product, NK-EVs are resistant to the influence of TME and TD-EVs, and can be designed for “off-the-shelf” use. In this systematic review, we examine the safety and efficacy of NK-EVs to treat various cancers in vitro and in vivo.

## 1. Introduction

According to the World Health Organization (WHO), the COVID-19 pandemic has claimed over 6.6 million lives globally [1]. Nonetheless, the incidence of infections and deaths has declined significantly with the availability of vaccines. Contradictorily, solutions for cancer treatment still remain obscure today. In 2021 alone, the WHO recorded 9.3 million global deaths from neoplasms [2]. Chemotherapy and radiotherapy are the standard treatment for cancerous tumors which cannot be removed entirely through surgery, despite the accompanying adverse effects [3,4]. These drugs and radiations damage the actively proliferating cancer cells and dividing normal cell indiscriminately [5,6,7,8], leading to deterioration in patient health by causing severe weight loss, poor skin conditions and hair loss (alopecia) [9,10]. The adverse effects of chemotherapy and radiotherapy in weakening patients’ health and immune system have yet to be resolved despite many years of refinement. In fact, criticisms from patients receiving these treatments discourages many from enrolling as their physical and mental wellbeing are at risk. The percentage of patients refusing to have or continue their chemotherapy is reported to be as high as 3–19% [11]. All these limitations demonstrate the importance of developing novel treatment for cancer which is highly specific and effective in eradicating the cancer cells without affecting the healthy cells.

Stem cells have been studied extensively in the field of regenerative medicine due to their excellent regenerative and anti-inflammatory properties [12]. They have been regarded as a viable solution for many age-related and chronic degenerative illnesses. This development has piqued the interest of utilizing stem cells for other therapeutic applications, including cancer immunotherapy. The development of cell-based immunotherapy can be achieved by establishing cell lines that express tumor selectivity and cytotoxic functions. Several prospective candidates that have been rigorously studied are natural killer (NK) cells, dendritic cells (DC), macrophages and T-lymphocytes [13,14,15,16]. These cells play critical roles in alerting the immune system (e.g., chemotaxis), arming other immune cells (e.g., antibody-dependent and cytokine-dependent activation) and exerting their cytotoxic effect against the cancer cells (e.g., receptor-mediated cell apoptosis). DC and CD4^+^ T-cells (T-helper cells) are important components of the immune system in that they serve as the moderators of various immune responses [17,18]. They produce molecular signals, e.g., cytokines and chemokines, to guide and activate other immune cells. Conversely, NK cells, macrophages and CD8^+^ T-cells (cytotoxic T-cells) are effector cells that exert a cytotoxic effect against the cancer cells [19,20,21,22].

The CD8^+^ T-cell’s mechanism of action is through the antibody-dependent cellular cytotoxicity (ADCC). ADCC is initiated by the binding of T-cell receptors (TCR) to the antigen presented by major histocompatibility complex-1 (MHC-1) [23]. NK cells also can eliminate the target cells via ADCC pathway through the cluster of differentiation-16 (CD16) or *FcγRIII* on its plasma membrane [24,25]. Upon activation, these cells will release cytolytic enzymes (e.g., caspases, granzymes (Gzm) and perforin (PFN)) to damage the target cells. Meanwhile, macrophages elicit antibody-dependent cellular phagocytosis (ADCP), which involves engulfment and degradation of the internalized cells via phagosome acidification [26,27]. Since mutational activities are frequent in cancer cells, antigen or peptide reconfigurations or direct inhibition of cytotoxic receptor (e.g., CA-125 inhibits *Fcy*-receptor) can disable ADCC/ADCP entirely [27,28,29,30,31]. In the event ADCC and ADCP are inhibited, the immune cells still can eliminate the cancer cells via the TRAIL and Fas signalling pathways. TRAIL and Fas receptors are “death receptors” which belong to the receptor family of tumor-necrosis factors (TNF) that induce programmed cell death when they are cross-linked by its ligands [32,33,34]. In addition, certain immune cells also secrete lytic granules containing Gzm and PFN that induce lysis or apoptosis of target cells via degranulation [35,36,37].

In spite of its potential, the progression of cell-based immunotherapy has been hampered by the limitations that it may cause life-threatening adverse reactions, laborious production procedure, limited efficacy against solid tumors due to insufficient trafficking and homing, and variable efficacy owing to immune cell inactivation by the tumors [38,39]. Recently, purification of cell biologics was made possible by the new innovations in isolation technique. By using these advanced isolation technologies, researchers have isolated the extracellular vesicles (EVs) produced by the immune cells and use them as cancer immunotherapy. In that regard, immune cells have been repurposed as biological manufacturers of these membrane-bound nanovesicles [40]. NK cells-derived EV (NK-EV) is being explored in many studies as it is rich in cytotoxic proteins, cytokines and miRNAs that selectively kill the tumor cells [41]. In the interest of this development, this systematic review was performed to determine the potential of NK-EVs as cancer immunotherapy based on data collected from the in vitro and in vivo experiments. EVs can be classified based on their biogenesis and size into exosomes (EXO), microvesicles (MVs) and apoptotic bodies [42,43]. In this review, NK-EVs are referring to the EXO and MVs produced by the NK cells.

## 2. Methods

Search keywords were selected using medical subject headings (MESH) available from PUBMED/MEDLINE. Common terms not available in MESH were also included in the search. Access to SCOPUS, PUBMED and Web of Science (WOS) was provided by the National University of Malaysia (UKM). All searches were filtered to either “research articles” or “journal articles” published in the last 5 years (2017–2022). All search registers were downloaded as bibliographies containing title, keywords and abstract. The bibliographies were labelled appropriately as source, date of access and results (e.g., PUBMED_210622_455 results). The bibliographies were then exported into Mendeley (Elsevier, The Netherlands). The merging of duplicates was performed automatically and manually. The first level of screening was based on relevance of title, abstract and keywords with topic of interest. After that, full-text research articles were downloaded for second level of screening. The final screening was performed following the inclusion and exclusion criteria as briefly described. Inclusion criteria: (i) NK cell, (ii) EV or EXO, (iii) cancer and (iv) controlled experimental studies. Exclusion criteria: (i) other immune cells, (ii) no EV or EXO, (iii) combined therapy and (iv) uncontrolled experimental study. Subsequently, the in vivo studies were reviewed for their risk of bias using SYRCLE’s ROB tool for animal studies. The protocol above has been registered (ID: CRD42022339710) in the international prospective register of systematic reviews (PROSPERO) by the National Institute for Health Research (United Kingdom).

We recorded a combined total of 1002 registers from three databases: PUBMED (330), SCOPUS (438) and WOS (234). A total of 340 registers were removed as a result of duplicate merging, leaving only 662 individual registers. The first process of screening removed 631 registers based on the title, abstract and keywords. This yielded 31 suitable articles for full-text retrieval. One article was not successfully retrieved since no English-text or translation was available. A total of 30 retrieved full-text articles were screened using the inclusion and exclusion criteria stated above. Finally, 18 individual registers containing either in vitro only, in vivo only or both in vitro and in vivo evidence were selected for data extraction and analysis. The article selection was performed by two authors (A.M.L.C and J.M.C.) and any disputes in article inclusion were resolved via discussion to reach mutual consensus. The article selection process is summarized in Figure 1.

## 3. Results and Discussion

### 3.1. NK-EVs Express Cytotoxicity against Various Human Cancer Cell Lines

A total of 17 studies performed in vitro experiments to examine the efficacy of NK-EVs against a panel of tumor cell lines, and all of them reported positive outcomes (Table 1). The NK-EVs were found to be effective against a wide range of cancer cell lines, shown in Figure 2 below, such as the brain [44,45,46,47,48,49,50,51], breast [44,45,47,49,50,51,52,53,54,55], blood [44,48,53,55,56,57], cervix [57], colon [50,51], liver [45,49,58], lung [57], ovary [51,54], pancreas [59], prostate [51], skin [51,60], stomach/gastric [50,60], and thyroid [45,49]. The dose of NK-EVs used in these studies ranged from low dose, between 0.3−5 µg, to median dose, between 10−25 µg, and high dose, between 40−100 µg. The NK-EVs were found to kill the cancer cells in a time- and dose-dependent manner [44,45,46,47,48,49,50,51,52,53,54,55,56,57,58,59,60]. Hence, higher cytotoxicity was recorded when the cancer cell lines were exposed to higher dose of NK-EVs for a longer period. Importantly, the NK-EVs showed selective cytotoxicity towards the cancer cell lines without affecting the viability of the healthy or normal cell lines tested. This evidence clearly showed that NK-EVs exhibited selectivity akin to their parental cells, as reported in the previous literature [61].

The impressive findings from the in vitro studies merely demonstrated the fundamental anti-cancer potential of NK-EVs without the interference of other biological systems. Thus, the use of mere in vitro data to show the anti-cancer potential of NK-EVs is totally insufficient, as the model does not replicate the physiological complexities that make up a living organism [62]. For more effective translation, a highly defined and complex model of pharmacodynamics (PD) and pharmacokinetics (PK) is needed, such as the one constructed by Bouhaddou et al. (2020) [63]. In term of cancer, the main debacle that has stumped the progression of many medical innovations is the existence of drug resistance, which is very difficult to replicate in vitro [64]. Tumor cells are known to interact with the surrounding cells and tissues, also known as tumor microenvironment (TME), to ensure their survival and the development of drug resistance. The TME provides important support needed by the tumor to thrive, including the secretion of signalling molecules by the proximally located cells, structural support and biochemical signals granted by the altered extracellular matrices (ECM), protection by the immunosuppressive local immune cells, nutrient supply and waste removal by the sprouting blood vessels, and a dynamic microenvironment (e.g., optimal temperature and pH level) that favor cancer cell growth, invasion and metastasis [65,66,67,68].

### 3.2. Tumor Microenvironment (TME): A Physicochemical Barrier against Immunotherapy

The TME is described as an extremely hostile ecosystem consisting of both physical and chemical components, as illustrated in Figure 3 [69,70]. Similar to all other cell types, cancerous cells are no exceptions to the innate ability to secrete EVs. In fact, tumor-derived extracellular vesicles (TD-EVs) are expressed in significant quantities, making them a plausible biological marker and progress indicator for cancer patients [71,72,73]. Cancer cells are also more self-sufficient than regular cells since they are capable of producing and responding to their own growth factors [74]. As demonstrated by El-Fattah Ibrahim et al. (2019), these tumor-enabling secretions can operate in an autocrine (own cells), paracrine (neighboring cells) and endocrine (distally located cells) manner [75]. TD-EVs were known to alter the surrounding microenvironment, making it hostile for immune cells and other healthy tissue, but their utility and mechanism of action were never fully comprehended. However, they do share a resemblance to anti-inflammatory secretions by supporting cell proliferation and angiogenesis, while inhibiting cell apoptosis, maturation or differentiation, and suppressing the recruitment of inflammatory-responsive cells (e.g., NK cells, macrophages, B and T lymphocytes) [76]. When healthy cells are replaced or destroyed, the surrounding environment is contaminated by the metabolic waste products, inflammasomes from cell lysis, and cell debris. Active glycolysis in malignant cells consumes the surrounding oxygen and releases acidic products (e.g., pyruvate and hydrogen ions) [77]. After depleting the surrounding oxygen, cancer cells will undergo anaerobic glycolysis, which contributes to further acidification via lactic acid production [78]. These manifest into the well-known acidic and hypoxic properties of TME, forming a chemical barrier that limits immune cell penetration.

What is also seemingly peculiar about TD-EVs is the manipulation of host cells, akin to a virus. TD-EVs are able to influence or reprogram the recipient cells to cooperate with tumors [79]. For example, TD-EVs can falsely trigger the anti-inflammatory machinery of the healthy cells to reduce the TME inflammation. On the other hand, death cells in TME will continuously release inflammatory particles. Hence, TME paradoxically houses both anti-inflammatory particles and pro-inflammatory particles. How these opposing factors co-exists and modulate the TME inflammation is yet to be fully understood. In addition, studies have shown that the circulating inflammatory bodies passively recruit immunoreactive cells to the tumor region [80,81,82,83]. The immune cells recruited to the TME will be modulated by the TD-EVs to deactivate its tumor cell killing function and/or to maintain its status quo via overstimulation of the inhibition:activation signal ratio [84]. At the same time, the aggregation of naïve/resting or inactivated immune cells interlocked by faulty ECMs obstructs any movement and acts as a shield against activated immune cells or medical interventions (e.g., chemotherapy drugs). Thus, the formation of this pseudo-barrier reinforces the physical impenetrability of the tumor, contributing to drug resistance.

TME and TD-EVs are known to work in harmony to disrupt NK cell function, thus, reducing the effectiveness of NK cell therapy. The acidic pH, as well as the immunosuppressive myeloid derived suppressor cells’ (MDSCs) and M2 macrophages’ presence in the TME, suppress the activation of NK cells, thereby compromising their cytotoxicity against a range of tumor cells [85]. Similarly, the TD-EVs were found to impair the function of NK cells through the transfer of multiple immunosuppressive factors (e.g., miRNAs and TGF-β) [86,87]. The existence of physical and chemical barriers in TEM paired with the inhibitory factors from TD-EVs render the NK cells and other immune cells almost entirely incompetent. Cellular release and uptake of EVs are known to increase in acidic environments [88]. Additionally, the pH-related stress also increases the EVs’ protein content and surface electrokinetic potential (zeta potential) [89]. The EVs with higher surface charges can bind strongly to the cell membrane, therefore increasing its internalization via receptor or non-receptor endocytosis. The higher EV absorption is not restricted to TD-EVs, but applicable to other EVs as well. This implies the possibility of higher absorption of NK-EVs and other immune cell-derived EVs by the cancer cells.

### 3.3. NK-EVs Show In Vivo Cytotoxicity in Tumor-Bearing Mice

The in vitro to in vivo extrapolation has seen numerous failures due to subpar experimental design and the presence of many limitations that yet to be addressed [62]. Thus, preclinical study needs to be performed to extrapolate animal data to humans despite that it is also unreliable due to species differences. Table 2 describes studies that assess the performance of NK-EV therapy using the in vivo model of tumor-bearing animals and the range of doses per administration [45,46,47,50,53,60,90]. The four studies shown in Figure 4 that performed intra-tumoral (IT) infusion reported significant suppression (*p* < 0.05) of tumor growth and lowered cancer cell viability in the neoplastic mass [45,53,60,90]. Similar results were reported in the seven studies that performed intravenous (IV) delivery of NK-EVs through the tail vein [45,46,47,49,50,58,90]. In all these studies, NK-EV infusion significantly (*p* < 0.05) reduced the bioluminescent intensity (BLI) signals and/or physical dimensions of the tumor mass compared to the untreated animals. The study by Zhu et al. (2018) performed a direct comparison between IT and IV administration routes in their animal models [45]. They found that IT administration is more effective (*p* < 0.05) compared to IV administration in shrinking the tumor mass. Similar results have been reported in previous studies examining the anti-tumor effect of immunotherapies in tumor-bearing mice [91,92]. These findings indicate that the IT route is likely to be more efficacious for NK-EV therapy but the IV route is still viable and may be used in specific situations when the IT route is not accessible.

The SYRCLE’s risk of bias tool adopted from Hooijman 2014 was used to assess the risk of bias of the selected studies (Figure 5) [93]. Generally, most of the elements evaluated showed an unknown risk of bias. While most of the studies demonstrated low risk of bias for some of the elements investigated, a few studies were found to have high risk. Among the nine studies reviewed, five studies (55.6%) explicitly mentioned randomization [45,46,47,50,58] but three studies (33.3%) did not [49,53,60]. A single study (11.1%) was deemed high-risk due to undisclosed sample size and allocation method [90]. The baseline was presented prior to treatment in 6 studies (66.7%) but ambiguous in the remaining three studies (33.3%). None of the studies disclosed information for allocation concealment, random housing, blinding of interventions or caregivers, random outcome assessment, and blinding of outcome assessment. Nearly all studies (*n* = 8, 88.9%) have unknown risk of bias for selective outcome reporting bias, with one study has high-risk of bias [50]. Five studies (55.5%) reported low risk of attrition bias and three (33.3%) had an unclear risk, with one (11.1%) having high-risk [50]. The unclear risk is mainly attributed to the inconsistency in the sample size of the study’s outcomes with the number of animals allocated per group. Failure to disclosure the number of animals (alive or dead) excluded from the analysis or parameter conducted were also took into account. All studies were at low risk of “other sources of bias” by disclosure of ethics approval, potential conflict of interest, or the collaborating parties and their roles.

### 3.4. NK-EVs Overcome the Chemical and Physical Barriers of TME

NK-EVs inherit the components and functions of NK cells to exert cytotoxicity against tumor cells in vitro and in vivo. NK-EVs not only express tumor-targeting ability but also possess homing and migratory properties via chemotaxis [45,46,47,50,58]. As depicted in Figure 6, NK-EVs’ framework is significantly more compact (30–200 nm) and they are also more biostable than NK cells [94,95]. These properties enable them to have improved infiltration and survival in the hostile TME. It has been demonstrated that NK-EVs express NKG2D, FasL and TRAIL just like the parent cells for receptor-mediated apoptosis. Lytic proteins such as PFN and Gzm also exist ubiquitously in NK-EVs to initiate caspase-dependent apoptosis pathway. All these functions show that the simple configuration of NK-EVs does not compromise their capability to eliminate cancer cells through the multiple cytotoxic pathways utilized by its parent cells. Instead, the NK-EVs might be more effective without the interference of the TME and TD-EVs that halt the production of PFN and Gzm in NK cells. It is believed that the anti-cancer potency of NK-EVs is mainly attributed by these killer proteins. However, the concentration and ratio of NK-EVs in relative to TD-EVs in the TME will likely determine if they will be able to exert anti-tumor effects. It is highly possible that the level of inhibitory signals induced by TD-EVs may overwhelm the activating signals of NK-EVs to prevent any cytotoxic action [84]. This means that the dose and delivery method of NK-EVs in vivo need to be critically evaluated to ensure their high bioavailability in the tumor in order to exert their cytotoxic function.

### 3.5. NK-EVs Are More Clinically Applicable than NK Cells

Results from the in vitro and in vivo experiments demonstrated that NK-EVs are an effective and safe immunotherapy. However, there are several foreseeable issues that need to be addressed in preparing this therapy for future clinical trials and registration. To begin, the functions of NK cells are distinguishable by their cell activation status [24,96,97]. Generally, both naïve and activated NK cells can secrete NK-EVs [98]. However, the activated NK cells are preferable for EV isolation due to their high cell number in in vitro culture. It has been reported that the NK-EVs derived from inactivated NK cells have lower amounts of cytotoxic proteins [46]. In a separate study, the authors found that EVs secreted by the NK92 cell line contain fewer cytotoxic proteins compared to the EVs secreted by in vitro-expanded NK cells [48]. Additionally, Shoae-Hassani et al. (2017) reported that NK cells exposed to neuroblastoma cells produced NK-EVs with higher cytotoxicity towards the tumor cells in vitro and in vivo compared to the NK-EVs derived from naïve NK cells [99]. Priming of NK cells with IL-15 was also reported to increase the cytotoxicity of NK-EVs [49]. These observations show that NK-EVs are highly heterogeneous depending on the cellular origin, culture environment and physiological status. Therefore, it is important to conduct experiments to the optimize the NK cell culture protocol in order to harvest NK-EVs with potent therapeutic potential. Importantly, the optimization should also consider increasing the yield of NK-EVs in order to reduce the cost.

Unlike the stem cell, donor-derived NK cells have variable and limited expansion potential in vitro [100]. Thus, efforts have been made to prepare immortalized NK cell lines, such as the NK92-MI from the American Type Culture Collection (ATCC, USA). From Table 1, eleven of the seventeen reviewed studies purchased this cell line to source their NK-EVs. Wu et al. (2019) and Aarsund et al. (2022) discovered little-to-no difference between the NK-EVs from NK92-MI cell line and those from freshly isolated peripheral blood-derived NK (PB-NK) cells [48,51]. Cryopreservation is widely used for long-term storage of NK cells. However, the cryopreserved NK cells have relatively poor viability albeit different freezing medium have been tested. Many studies have reported cell viability less than 10% after revival [101]. More alarming is that the survived cells ceased proliferating and failed to exert any cytotoxic functions [102]. Failure of cell cryopreservation will definitely hamper the clinical translation of NK-EV therapy as it will limit the potential of producing them on a large scale and consistently. At the moment, new donor-derived NK cells are used to prepare the NK-EVs, which leads to high batch-to-batch variation. Furthermore, it will also increase the production cost and lead time.

Manual cell culture protocol using tissue culture flask is prone to contamination, technical errors and high batch-to-batch variation [103,104]. To overcome these limitations, efforts should be made to shift the NK cell expansion to a bioreactor platform which is highly automated, allows more control over the culture environment (e.g., glucose concentration, oxygen concentration and pH) and requires less manpower [105]. Usage of a highly efficient bioreactor system might also reduce the cost of production [44]. Not least, it is important is to optimize the EV isolation protocol, which should allow quick and reliable isolation of EV subpopulation.

Last but not least, long-term storage and off-the-shelf availability are highly desirable for NK-EV therapy. To achieve this, researchers need to device a storage condition that can preserve EV stability over a long period of time. As an acellular product, NK-EVs can be kept chill (4 °C) for immediate use or frozen (−20 °C or −80 °C) up to 12 months with acceptable loss of proteins and RNAs [106,107,108,109,110]. More importantly, the NK-EVs retain their anti-tumor effect. For easier storage and transportation, NK-EVs could be lyophilized. Even though lyophilization has been widely explored for storage of stem cell-derived EVs, it is still unknown whether lyophilization can be used to preserve the NK-EVs.

### 3.6. Defining Nomenclature, Isolation Technique, QA/QC Methods and Biomarkers for NK-EVs

In the very beginning, EVs were first thought as a mechanism of disposing cellular waste products [111]. Later, studies found that these minute vesicles are rich in lipids, proteins, nucleic materials (e.g., DNA, RNA, mRNA, miRNA) and other biologically active molecules [112]. On their surface, they express different molecules and proteins adopted from parent cell membrane during the budding or exocytosis process. These physical hallmarks contribute to the diverse functions and heterogeneous properties of EV molecules. Size is commonly used to classify EVs [113]. The largest EVs are the apoptotic bodies (1000–5000 nm), followed by microvesicles (50–1000 nm), with exosomes (30–150 nm) being the smallest [114]. The term “exosome” is often incorrectly used interchangeably with “extracellular vesicle” to describe the small EV preparation [42]. This is because most of the studies isolate the EVs based on their size instead of the specific phenotype that represented their unique biogenesis pathway. Every EV subset has a specific protein phenotype and molecular content. Thus, the Minimal Information for Studies of Extracellular Vesicles (MISEV) 2018 guidelines have recommended the nomenclature “extracellular vesicles” to describe the small membrane-bound particles secreted by the cells. It is crucial to establish a uniform and specific standard for appropriate evaluation of different NK-EV preparations as it would allow researchers to determine the strengths and weaknesses of any modifications or novelties introduced.

Based on Table 3 and Figure 7, the most commonly used EV isolation method is the ultracentrifugation (UC) [44,45,50,52,53,54,55,57,58,60,90]. UC is often used in combination with other EV isolation techniques to improve the purity [45,50,53,58,60]. Alternative isolation techniques used in the reviewed studies include density gradient centrifugation (DGC) [45,49,50,53,60], polyethylene glycol-8000 (PEG8000) precipitation [44,48,53], differential centrifugation (DC) [47,56,58] and size exclusion chromatography (SEC) [46,49,51]. Generally, the existing techniques are unable to achieve both high recovery and high specificity. With the rise in interest in EV-related therapeutics especially mesenchymal stem cell-derived EVs (MSC-EVs) and EV-based immunotherapy, it is vital to develop a novel isolation method which can give high recovery and high purity as well as can be easily scaled up [42,113]. Currently, tangential flow filtration (TFF) is being revisited and improved for more efficient recovery of EVs [58,115,116]. Watson et al. (2018) utilized the combined platforms of TFF and SEC to isolate MSC-EVs and reported that TFF is a reproducible and scalable technique to recover clinical-grade EVs compared to other known techniques [117].

According to MISEV 2018, EVs should be categorized based on size and biochemical composition (e.g., CD9, CD81 and TSG101) [118,119]. Most of the selected articles in this study used small NK-EVs in the range 50–200 nm [44,45,46,47,49,51,52,53,55,56,57,58,59,60,90]. Only Jiang et al. used a subset of EVs that are larger in diameter, i.e., 205.6 ± 29.65 nm [54]. To prevent misreport or misrepresentation of each author’s work, only text-based evidence of the NK-EV sizes were recorded since not all reviewed studies had specifically disclosed the diameter, mean or mode of their NK-EVs. The frequently used assessment tools for EVs in these studies includes nanoparticle tracking analysis (NTA) to examine the size [44,45,46,47,49,51,53,55,56,57,58,90], transmission electron microscopy (TEM) to study the size and morphology [44,45,46,47,49,51,52,53,54,55,58,60], Western blot (WB) [44,45,46,47,48,49,50,51,52,53,56,57,58,60,90] and flow cytometry (FC) [49,54,55,56,57,59,60] to determine the biochemical markers, and bicinchoninic acid (BCA) protein assay [47,50,51,53,54,57,60] to quantify the protein concentration. The biochemical markers consistently tested to characterize the EVs were CD63 [44,45,47,49,50,51,52,53,54,55,56,57,58,59,60,89], CD81 [46,51,56,57,89], ALIX [45,46,47,49,52,53,58,60,90] and TSG101 [46,47,51,52,53,54,56,59,90]. Furthermore, specific markers, i.e., Fas/FasL or Fas/TRAIL [45,48,50,51,54,57,58], granulysin (GNLY) [44,48,52,57], granzymes A (GzmA) and/or B (GzmB) [44,46,48,49,51,53,54,56,57], and PFN [44,45,46,48,49,51,53,54,56,57,59,60] were used to prove that the EVs originated from NK cells. The characterization of NK-EVs using the above techniques is crucial to verify the NK-EVs, as well as to compare and standardize the NK-EV research.

### 3.7. Future Considerations of NK-EVs as a Tool for Immunotherapy

Table 4 summarizes the established benefits and challenges of NK cells and NK-EVs for immunotherapy. However, the discussion going forward shall emphasize on improving the EV-based immunotherapy as it has yet to be thoroughly explored but already exhibiting significant advantages compared to the former. One of the main benefits of cell-based and EV-based therapies is that they are minimally invasive medical procedures [120]. They are safer alternatives with potentially fewer procedural complications to the high-risk populations such as infants and elderly. In contrast, surgical treatment can lead to many serious complications, including induced hemorrhage, post-operative sepsis and extended rehabilitation [121,122]. Selection of administration route or delivery technique is critical to ensure the safety and efficacy of a therapy [123]. Although IT administration would eliminate any concerns over the homing and migration ability of NK-EVs, the IV administration will be especially useful for regions with poor access or penetrability (e.g., brain) [124]. In actuality, a larger dose is typically needed for IV route to compensate for the accumulated loss of drugs due to unwanted distribution at the other tissues as well as metabolism and excretion primary at the liver and kidney [125,126,127]. Nonetheless, the continual IV administration of NK-EVs after the tumor elimination could be favourable to ensure complete clearing of tumor cells to minimize the risk of relapse [128,129]. To date, it is still unclear on the optimal dosage regime of NK-EVs. More research is needed to determine the ideal frequency, dosage and interval of NK-EV treatment.

Efforts are also needed to improve the efficacy of NK-EV production in order to reduce the cost. Owing to its niche role, NK cells have high metabolic activity and need specific cytokines to stimulate its proliferation. The commonly used cytokines to expand NK cells are IL-2, IL-15, IL-18, and IL-21. The cytokines can be provided by the peripheral blood mononuclear cells (PBMCs) that serve as the feeder cells [130]. Alternatively, the culture medium can be supplemented with commercially available human recombinant cytokines to maintain the NK cell culture. Although the use of recombinant cytokines is simpler as no purification step is needed to remove the contaminating PBMCs and safer as it is free from PBMC contamination, the cost of recombinant cytokines is very high. The high concentration and cost of recombinant cytokines will significantly escalate the production cost. On top of that, the requirement of using GMP-grade or clinical-grade raw materials and consumables, skilled manpower, stringent quality assurance and quality control (QA/QC), expansive equipment for scalable production, and the use of a cGMP-certified production facility will escalate the treatment cost to unaffordable scales for majority of the patients [131]. This, in turn, will affect the clinical translation and commercial viability of NK-EV therapy.

## 4. Conclusions

NK-EVs show promising tumor cytotoxicity with no adverse or side effects in vitro and in vivo. NK-EVs are not cytotoxic to the normal or healthy cells and can increase the life span of tumor-bearing animals. Thus, this novel therapy has great clinical potential. In addition, NK-EVs can circumvent some of the limitations of NK cell therapy, such as impediment of NK cell functionality by the TME and TD-EVs. In fact, NK-EVs would see potential benefits due to the increased cellular uptake in the TME. Although IT administration is superior, the IV administration will be meaningful for “difficult-to-reach” tumors and high-risk populations. Additionally, NK-EVs can be readily available off-the-shelf as they are more stable and easier to store compared to NK cells. Nonetheless, NK-EV therapy and its production procedure need further investigation to improve its safety and efficacy as well as to increase the yield and lower the production cost.

## Figures and Tables

**Figure 1 ijms-24-04026-f001:**
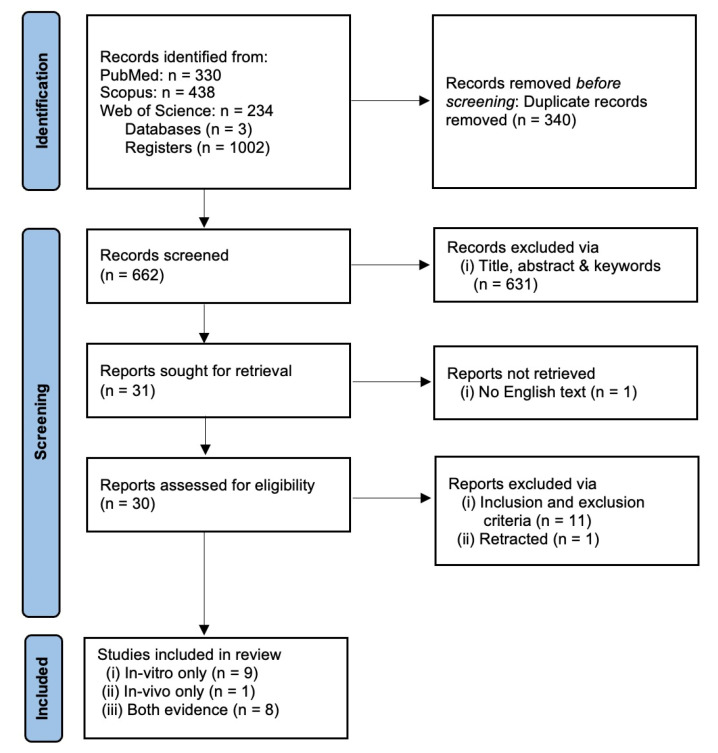
PRISMA flow chart for systematic review.

**Figure 2 ijms-24-04026-f002:**
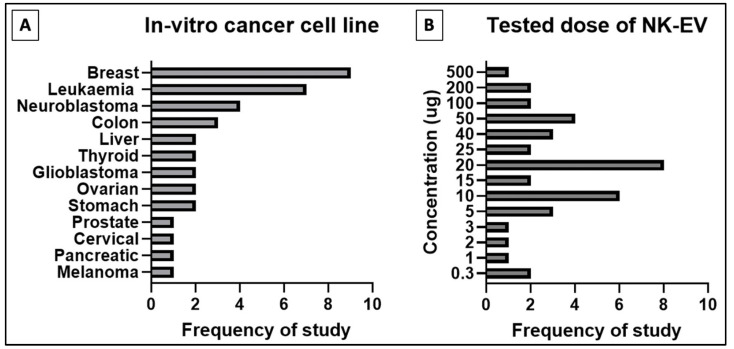
Distribution of in-vitro studies for (**A**) type of cancer cell line studied and (**B**) tested doses of NK-EVs for cancer cell cytotoxic assay.

**Figure 3 ijms-24-04026-f003:**
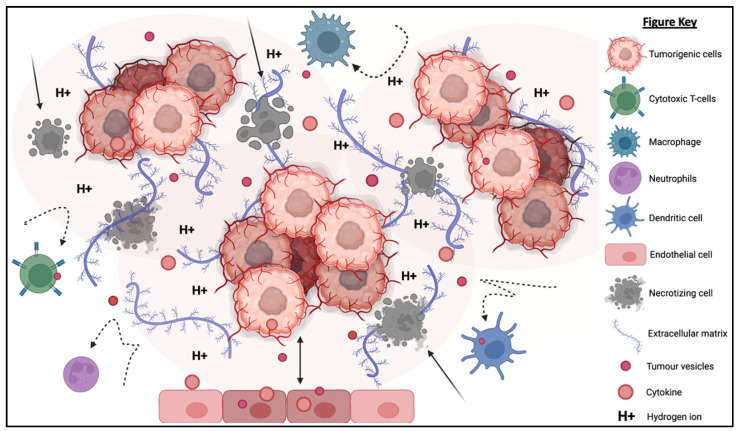
The tumorigenic cells are centered around faulty extracellular matrices, necrotic bodies, and a hypoxic and acidified microenvironment. This tumor microenvironment discourages immunoreactive cells from responding or otherwise entering the tumor to eradicate the tumor cells. At the same time, the hypoxic and acidic environment will induce cell death and exacerbate the inflammation. Moreover, tumor-derived extracellular vesicles will reprogram the host’s cells (e.g., endothelial cells) to support tumor development via increasing the secretion of growth factor, pro-angiogenic and anti-inflammatory cytokines.

**Figure 4 ijms-24-04026-f004:**
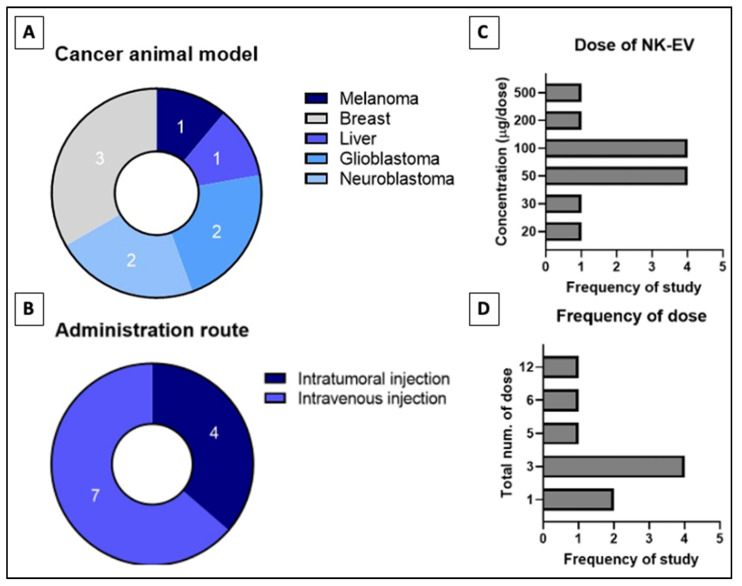
Distribution of in-vivo studies for (**A**) type of cancer studied, (**B**) administration route of NK-EVs, (**C**) tested dose of NK-EVs per dose and the (**D**) frequency of dose administered throughout study phase. Numbers indicated in (**A**,**B**) refers to the frequency of reviewed studies that fall into the categories.

**Figure 5 ijms-24-04026-f005:**
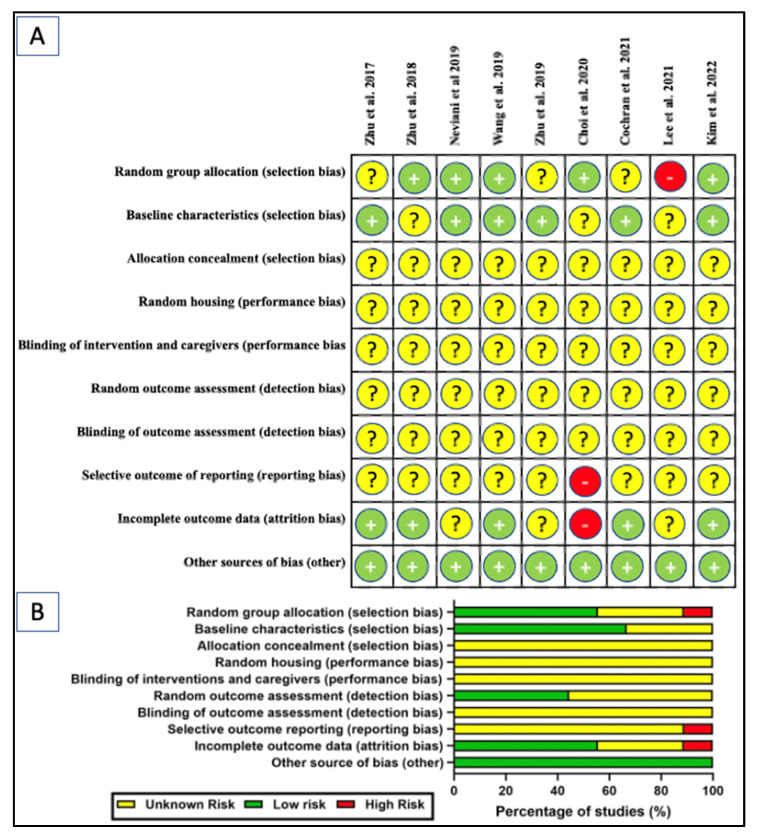
SYRCLE’s risk of bias tool: (**A**) scoring of individual studies; (**B**) overall scores presented as percentage of studies for in-vivo studies [45,46,47,49,50,53,60,90]. Symbols: yellow “?” indicates unclear risk; green “+” indicates low risk; and red “-” indicates high risk of bias.

**Figure 6 ijms-24-04026-f006:**
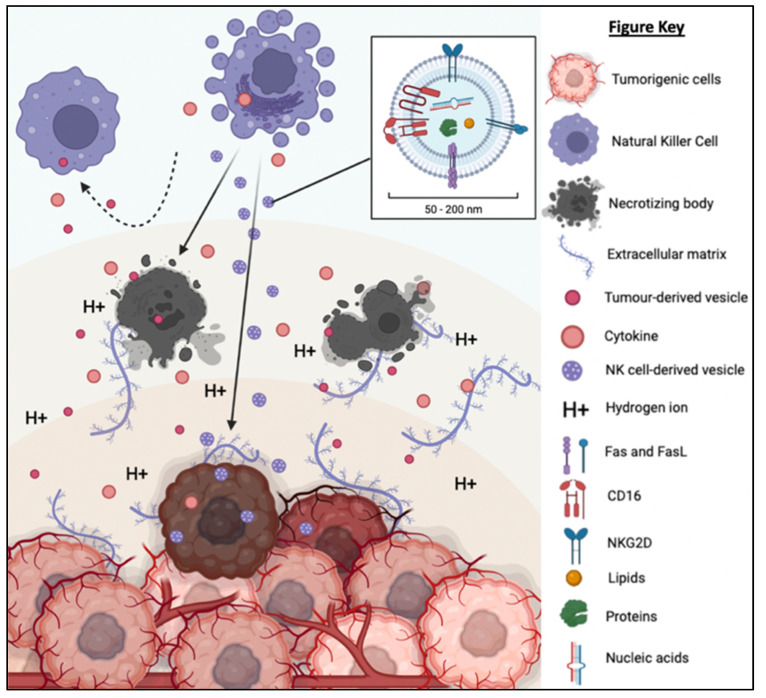
TD-EVs suppress the recruitment and migration as well as reduce the proliferation, survival and cytolytic function of NK cells. On the other hand, NK-EVs secreted by the NK cells are small and diligent enough to overcome the physical and chemical barriers of TME to reach and exert its cytolytic effect on the tumor cells.

**Figure 7 ijms-24-04026-f007:**
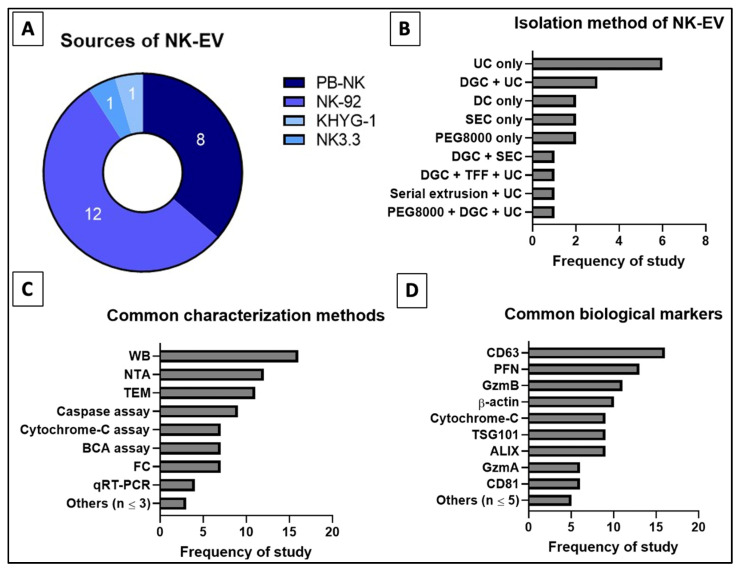
Distribution of reviewed studies for (**A**) source, (**B**) isolation techniques (single or combined), (**C**) common characterization assays and (**D**) common biological markers for NK-EVs. Numbers indicated in (**A**) refer to the frequency of reviewed studies that fall into the categories.

**Table 1 ijms-24-04026-t001:** In vitro evidence demonstrating cytotoxicity of NK-EVs against various cancer cell lines.

Study	Type of NK-EV	Dose of NK-EV	Tested Cancer Cell Lines	Key Findings
Jong et al., 2017 [44]	NK-EVs from PB-NK of healthy donors	20 or 40 µg	Human acute lymphoblastic leukemia (NALM-6; SUPB15), neuroblastoma (CHLA-136; CHLA-255) and breast cancer (MCF-7)	NK-EVs increased apoptosis of all tumor cell lines in a time- and dose-dependent manner.
Zhu et al., 2017 [60]	NK-EXO from human NK cell line (NK92-MI)	5 or 20 µg	Mouse melanoma (B16F10); Human gastric carcinoma (SNU484) and colon cancer (HCT-5)	All cell lines showed lower cell viability in a time and dose-dependent manner. NK-EXO did not illicit any response in healthy cells. NK-EXO induced cancer cell apoptosis through PFN and Gzm as well as activation of Fas/FasL pathway.
Zhu et al., 2018 [45]	NK-EXO and NK-EM from human NK cell line (NK92-MI)	10, 20 or 30 µg/mL	Human glioblastoma (D54), breast carcinoma (MDA-MB-231), anaplastic thyroid cancer (CAL-62) and hepatic carcinoma (HepG2)	NK-EM showed greater anti-tumor properties compared to NK-EXO. Both treatment groups reduced BLI signal intensity for all tested tumor cell lines in a time and dose-dependent manner.
Neviani et al., 2019 [46]	miR-186 enriched NK-EVs from PB-NK of healthy donors	A series of 2-fold dilutions starting from 4 × 10^11^ particles/mL	Human MYCN-amplified (CHLA-136 and LAN-5) and non-amplified (CHLA-255) neuroblastoma	IL-15 treated NK cells and its exosomes successfully halted growth of neuroblastoma cell lines. Inactivation of NK cells via TGFβ1 did not affect the functions of the secreted exosomes despite downregulation of cytotoxic proteins. miR-186 delivery via NK-EXO reduced expression of tumor escape oncogenes, i.e., MYCN, AURKA, TGFβ1R and TGFβ2R.
Sun et al., 2019 [59]	miR-3607-3p enriched NK-EVs from PB-NK of healthy donors	−	Human pancreatic cancer (MIA PaCa-2 and PANC-1)	NK-EVs inhibited growth, migration and invasive properties of both cancer cell lines.
Wang et al., 2019 [47]	NK-EXO and NN/NK-EXO from PB-NK of healthy donors	10, 20 or 40 µg	Human breast cancer (MDA-MB-231) and neuroblastoma (CHLA-255)	NK-EXOs reduced the viability of both tumor cell lines in a dose-dependent manner. NN/NK-EXOs showed higher levels of tumor cytotoxicity compared to NK-EXOs.
Wu et al., 2019 [48]	NK-EVs from PB-NK of healthy donors and human NK cell line (NK92-MI)	40 µg/100 µL	Human acute lymphoblastic leukemia (SUPB15) and neuroblastoma (CHLA255)	NK-EVs were cytotoxic towards both cancer cell lines. PFN, GzmA, GzmB and GNLY were found to work collectively to induce tumor cytotoxicity. NK-EVs were able to enter caspase-dependent and independent pathways, highlighting the flexibility of NK-EVs to access multiple cytotoxic pathways.
Zhu et al., 2019 [49]	NK-EVs from human NK cell line (NK-92MI) with or without IL-15 treatment	5, 10 or 15 µg	Human breast cancer (MDA-MB-231), anaplastic thyroid cancer (CAL-62) and glioblastoma (U87/MG)	Both NK-EV_IL-15_ and NK-EVs displayed dose- and time-dependent cytotoxicity against all cancer cell lines. NK-EV_IL-15_ had greater cytotoxic effect compared to NK-EVs.
Choi et al., 2020 [50]	NK-EVs from PB-NK of healthy donors	2, 5, 10 or 20 µg	Human hepatocarcinoma (HEPG2), colon cancer (SW-620), stomach cancer (MKN-74), breast cancer (MCF-7) and brain cancer (T98G).	NK-EVs were cytotoxicity against all cancer cell lines.
Di Pace et al., 2020 [56]	NK-EXO from IL-2 or IL-15 stimulated PB-NK of healthy donors	5, 20 or 50 µg/100 µL	Human childhood B acute lymphoblastic leukemia (NALM-18) and erythroleukemia (K562)	NK-EXOs exerted cytotoxicity against both cancer cell lines in a dose-dependent manner.
Han et al., 2020 [52]	NK-EXO and PTX-NK-EXO from human NK cell line (NK92-MI)	40 µg/mL of NK-EXO or 15 µg/mL of PTX-NK-EXO	Human breast cancer (MCF-7)	NK-EXOs were as cytotoxic as PTX against the breast cancer cell line. NK-EXOs showed great potential as cancer drug carriers with PTX-NK-EXOs showed the highest tumor cytotoxicity.
Cochran et al., 2021 [53]	NK-EVs from human NK cell lines (NK3.3 and NK92-MI)	1, 10, 25, 50 or 100 μg/mL	Human T cell leukemia (K562 and JURKAT) and breast cancer (HEK293, MCF-7 and MDA-MB-231)	NK-EVs showed cytotoxicity against both cancer cell lines in a time- and dose-dependent manner. NK-EVs did not trigger apoptosis in normal cells, i.e., HEK293 and PB- and CB-derived lymphocytes.
Enomoto et al., 2021 [57]	NK-EV from human NK cell line (NK92-MI)	0.3, 1 or 3 μg	Human T cell leukemia (K562 and JURKAT), lung (A549) and cervical cancer (HELA)	NK-EVs exerted time- and dose-dependent cytotoxic towards all the cancer cell lines albeit the dosage is much lower compared to other studies.
Jiang et al., 2021 [54]	NK92-EXO from human NK cell lines (NK92-MI and NK92-hIL-15) cultured in either normal or hypoxic condition	25 or 50 μg/mL	Human breast cancer (MCF-7) and ovarian cancer (A2780)	NK cells in hypoxic culture (24 and 48 h) doubled its EV production. Hypoxic and normoxic NK-EVs demonstrated similar degree of cytotoxic towards the cancer cell lines.
Kaban et al., 2021 [55]	NK-EXO from human NK cell line (NK92-MI) overexpressing BCL-2 siRNAs	200 µg/mL	Human T cell leukemia (K562 and MEC1) and breast cancer (HEK293T, SKBR3, MCF-7, MCF-10A, T-47D and MDA-MB-231)	Transduced cells produced EVs enriched with BCL-2 siRNAs. Modified NK-EVs showed greater cytotoxicity against the breast cancer cell lines but have no effect on the normal cells.
Aarsund et al., 2022 [51]	NK-EVs from IL-15 or IL-12/15/18-stimulated PB-NK of healthy donors and human NK cell line (NK92 and KHYG-1)	20 µg	Human colon cancer (HCT116 and HCT-15), prostate cancer (DU145 and PC3), breast cancer (SK-BR-3 and T-4D7), ovarian cancer (OVCAR-3), leukemia (KHYG-1), melanoma (WM9) and glioblastoma (U87).	NK-EVs isolated from IL-15 and IL-12/15/18-stimulated NK cells and NK-92 cells were able to kill the cancer cells in 2D and spheroid cultures. KHYG-1 EVs showed no tumor cytotoxicity in both in vitro models.
Kim et al., 2022 [58]	NK-EXO from human NK cell line (NK-92)	10, 20, 50, 100, 200 or 500 µg	Human hepatocellular carcinoma (Hep3B, HepG2 and Huh7)	NK-EXOs had a cytotoxicity towards Hep3B compared to HepG2 and Huh7 cells in a dose-dependent but not time-dependent manner.

**Table 2 ijms-24-04026-t002:** In vivo evidence demonstrating the antitumor potential of NK-EVs.

Study	Type of Cancer and Animal Model (*n* = Sample Size per Group)	Dosage and Method of Administration	Key Findings
Zhu et al., 2017 [60]	B16F10/effluc cells (1 × 10^5^ cells/100 µL) were subcutaneously injected into the right thigh of pathogen-free 6-week-old female C57BL/6 mice (*n* = 6)	20 µg/100 µL of human NK-EXO via IT	Tumor was effectively reduced (3.5 folds) after 2 to 5 days. In vivo and ex vivo BLI confirmed reduced signal intensity and reduced tumor mass in the treatment group.
Zhu et al., 2018 [45]	D54 cells (5 × 10^6^ cells/100 µL) were subcutaneously injected into pathogen-free 6-week-old female BALB/c mice (*n* = 5)	100 µg/150 µL (via IV) and 30 µg/50 µL (via IT) of human NK-EM, 3 times at intervals of 3 days	Both treatments reduced the tumor mass with IT route showed greater reduction.
Neviani et al., 2019 [46]	CHLA-136-Fluc cells (1 × 10^6^ cells) were intra-renally injected into left kidney of 4- to 8-week-old female and male NSG mice (*n* = 5 − 10)	1 mg/kg/d of miR-186 enriched human NK-EVs via IV 3 times per week	BLI signal intensity of tumor cells and weight of kidneys were decreased in the animals received miR-186-enriched NK-EVs compared to the control group. The treated animals showed improved survival rate.
Wang et al., 2019 [47]	CHLA-255-luc cells (1 × 10^7^ cells/500 µL) were injected via IV into specific pathogen-free, 6-week-old female NOD/SCID mice (*n* = 3)	100 µg of human NK-EXO with or without NN let-7a loaded polyamidoamine dendrimer via IV route	NN/NK-EXOs showed better homing efficacy and greater suppression of tumor growth compared to the NK-EXOs.
Zhu et al., 2019 [49]	U87/MG/F cells were administered into specific pathogen-free, 6-week-old female BALB/c nude mice (*n* = 15)	50 µg of human NK-EVs or 50 µg of human NK-EVs_IL-15_ via IV for 5 times at intervals of 2 days	Both treatments significantly reduced BLI signals and tumor weight compared to the control group. NK-EVs_IL-15_ were significantly more effective compared to NK-EVs. Both treatments did not elicit toxic response in tumor-bearing animal model.
Choi et al., 2020 [50]	MCF-7 cells (3 × 10^6^ cells) were injected subcutaneously into the right flank of 5-week-old, female athymic nude mice (*n* = 4)	50 µg/100 µL of human NK-EVs via IV 3 times weekly	The tumor dimension and mass reduced significantly compared to the control group after 2 weeks.
Cochran et al., 2021 [53]	GFP-expressing MDA-MB-231 cells (2 × 10^6^ cells) were injected into the 4th mammary fat pads of female athymic nude mice (*n* = 4–5)	50 µg of human NK3.3-EVs via IT, 7 times at intervals of 3 to 4 days	The NK3.3-EV treated animals showed a higher number of dead cells both histologically and in TUNEL assay.
Kim et al., 2022 [58]	Hep3B cells (1 × 10^7^ cells/100 µL) was subcutaneously (right back) or (2 × 10^6^ cells/50 µL) orthotopically (liver) xenografted into 6-week-old male BALB/c nude mice (*n* = 5)	50, 100, 200 or 500 µg of human NK-EXO via IV 6 times at intervals of 2 days	NK-EXO exerted migratory and targeting ability to inhibit tumor growth in dose-dependent manner in both subcutaneous and orthotopic animal models.
Lee et al., 2021 [90]	Canine REM134 cells (1 × 10^4^ cells) xenografted into mammary fat pad of BALB/c nude mice via IT route (*n* = n/a)	100 µg of canine NK-EXO via IT once and IV twice per week for 6 weeks	NK-EXO suppressed tumor growth and reduced the expression of tumor-associated markers.

Abbreviations: BLI—Bioluminescent imaging; IT—Intra-tumoral; IV—Intravenous; NK-EM—Natural Killer cell-derived exosome-mimetic vesicle; NSG—NOD-SCID gamma mice; PBS—Phosphate buffered saline.

**Table 3 ijms-24-04026-t003:** Method of isolation, characterization and functional analysis of NK-EVs.

Study	Isolation Method	Size Range or mean (±SEM) (nm)	Size Peak or Mode (nm)	Methods of Assessment	Identification Markers	Functional Markers
Jong et al., 2017 [44]	PEG8000	155 ± 5.9	120 ± 6.4	Caspase assay, NTA, TEM, WB	CD63, FN, GNLY, GzmA, GzmB, PFN	CD56, Cytochrome-C, Rab5A
Zhu et al., 2018 [45]	^a^ Serial extrusion + UC or ^b^ UC + DGC	^a^ 100–150 ^b^ 100–120	^a^ 99.2 ± 21.5 ^b^ 118 ± 33.1	Caspase assay, Cellular uptake assay, Inhibition assay, NTA, TEM, WB	ALIX, β-actin, CD63, Cytochrome-C, Fas, GM-130	AKT, ERK, PFN, p-AKT, p-ERK
Neviani et al., 2019 [46]	SEC	122.2 ± 1.3	92.5 ± 1.2	NTA, TEM, WB	ALIX, β-actin, CANX, CD81, FN, HSP70, TSG101	AURKA, GzmA, GzmB, MYCN, PFN, TGFBR1, TGFBR2
Wang et al., 2019 [47]	DC	100	−	BCA assay, DLS, NTA, TEM, WB	ALIX, CD63, TSG101	CD47, CXCR4, Cytochrome-C
Wu et al., 2019 [48]	PEG8000	−	−	Caspase assay, Cytochrome-C assay, ELISA, WB	β-actin, Cytochrome-C FasL, GzmA, GzmB, GNLY, PFN	−
Zhu et al., 2019 [49]	SEC + DGC	NK-EVs: 106.9 ± 21.6 NK-EVs_IL-15_:118.2 ± 20.3	−	BLI, Caspase assay, FC, MTT assay, NTA, TEM, WB	ALIX, β-actin, Calnexin, CD63, Cytochrome-C, GM-130	Membrane-FasL, Cytoplasm-FasL, PFN, GzmB
Choi et al., 2020 [50]	DGC + UC	−	−	2-DE proteome analysis, Antibody blocking assay, BCA assay, WB	CD40L, CD49, CD51, CD63, Integrin α1, Integrin α3, Integrin β1, L-selectin	Apo A-IV, Apo E, β-actin, DR4, DR5, DNAM-1, Fas, FasL, FGB, FGG, FN, HSP90 α/β, IFN-γ, IL-6, L-plastin, NKG2D, NKP44, NKP46, TRAIL, TNF-α, VCP
Aarsund et al., 2022 [51]	SEC	60–125	−	BCA assay, LC-MS/MS, NTA, TEM, WB	CD63, CD81, FasL, GzmB, PFN, TSG101	DNAM-1, NKG2D, NKP46, NKP30
Han et al., 2020 [52]	UC	80–110	−	DLS, HPLC, TEM, qRT-PCR, WB	ALIX, CD63, TSG101	Bax, Bcl-2, β-actin, Cas-3,
Cochran et al., 2021 [53]	PEG8000 + DGC + UC	188.6 ± 2.7	133.4 ± 8.0	BCA assay, Caspase assay, GO, LC-MS/MS, qPCR, NTA, TEM, WB	ALIX, Annexin V, β-actin, CD9, CD63, HSP70, HSP90, LAMP1, NKLAM, TSG101	DNAM1, GNLY, GzmA, GzmB, ICAM1, MHC-I, MHC-II, PFN, VCAM1
Jiang et al., 2021 [54]	UC	50–200	205.6 ± 29.65	BCA assay, FC, TEM, WB, Wound healing assay	CD63, FasL, GAPDH, GzmB, PFN, TSG101	−
Kaban et al., 2021 [55]	UC	115.8–128.9	−	Caspase assay, FC, NTA, TEM, Immunogold staining, qRT-PCR	CD56, CD63	7-AAD, Annexin V, Bcl-2, Cas 3/7, Cas-9, Cytochrome-C, TMRE
Di Pace et al., 2020 [56]	DC	135.9 ± 0.5	88 ± 1.3	Bradford assay, Caspase assay, Cellular uptake assay, ELISA, FC, NTA, WB	CANX, CD63, CD81, TSG101	CD3, CD16, CD19, CD56, CD69, DNAM1, GzmA, GzmB, IFN-γ, LFA-1, NKP44, NKG2D, PFN, PD-1
Enomoto et al., 2021 [57]	UC	−	148.2	BCA assay, FC, Migration assay, miRNA profiling (small RNA-seq), MS, NTA, GO, qRT-PCR, WB	α-tubulin, β-actin, CD63, CD81, Cytochrome-C	CD226, FasL, GNLY, GzmA, GzmB, GzmH, PFN, TRAIL
Kim et al., 2022 [58]	UC + DC + TFF	^a^ 106.1 ± 71.5 ^b^ 128.5 ± 33.3	−	Caspase assay, Cellular uptake assay, ^a^ DLS, LDH assay, ^b^ NTA, TEM, WB	ALIX, CD63, CD81, GzmB, PFN, TRAIL, FasL	β-actin, Cas3, Cas7, Cas8, Cas9, Cytochrome-C, PARP, p-AKT, p-ERK1/2
Sun et al., 2019 [59]	UC	50–200	−	FC, Migration and invasion assay, SEM, qRT-PCR	ACTIN, CD63, TSG101	IL-26, mir-3607-3p
Zhu et al., 2017 [60]	DGC + UC	100–150	−	BCA assay, Caspase assay, ELISA, FC, TEM, WB	ALIX, CD63, β-actin, GM-130	Annexin V, Cytochrome-C, FasL, PFN, p38, TNF-α
Lee et al., 2021 [90]	UC	136.6 ± 9.4	−	NTA, WB	ALIX, CD63, CD81, HSP70, TSG101, GzmB, PFN	Bax, Bcl-xL, Bmi-1, CD133, IL-1β, IL-6, MDR, MMP-3, p53, PCNA, TNF-α, VEGF

Symbol: “−” indicates data not specified or available by authors. Abbreviations: 2-DE—2-dimensional gel electrophoresis; 7-AAD—7-aminoactinomycin D; AKT—protein kinase B; ALIX—ALG-2-interacting protein X; APO—apolipoprotein; Bax—Bcl-2-associated X protein; BCA—bicinchoninic acid assay; BCL2—B-cell lymphoma 2 gene; CANX—calnexin; Cas—caspase; CXCR4—C-X-C chemokine receptor type 4; DC—differential centrifugation; DGC—density gradient centrifugation; DLS—dynamic light scattering; DNAM1—DNAX accessory molecule-1; DR4—death receptor 4; ELISA—enzyme-linked immunosorbent assay; FC—flow cytometry; FGB—fibrinogen beta chain; FGG—fibrinogen gamma chain; FN—fibronectin; GFP—green fluorescent protein; GM-130—golgi matrix protein; GNLY—granulysin; GO—gene ontology; GzmA—granzyme A; GzmB—granzyme B; HPLC—high performance liquid chromatography; HSP70—70-kilodalton heat shock proteins; ICAM1—intercellular adhesion molecule 1; IFN—interferon; LAMP1—lysosomal-associated membrane protein 1; LC-MS/MS—liquid chromatography–tandem mass spectrometry; LFA-1—lymphocyte function-associated antigen 1; MHC—major histocompatibility complex; MS—mass spectrometry; NKG2D—natural killer group 2D; NKP44—natural killer cell P44-related protein; NN—nanoparticles; NTA—nanoparticle tracking analysis; PARP—poly [ADP-ribose] polymerase; PD1—programmed cell death 1; PEG8000—polyethylene glycol 8000; PFN—perforin; qPCR—quantitative polymerase chain reaction; qRT-PCR—quantitative real-time polymerase chain reaction; SEC—size exclusion chromatography; SEM—scanning electron microscopy; TFF—tangential flow filtration; TEM—transmission electron microscopy; TMRE—tetramethylrhodamine, ethyl ester; TNF—tumor necrosis factor; TRAIL—TNF-related apoptosis inducing ligand; TSG101—tumor susceptibility gene 101; UC—ultracentrifugation; VCAM-1—vascular adhesion molecule 1; VCP—valosin-containing protein; WB—Western blot.

**Table 4 ijms-24-04026-t004:** Summary of pros and cons of NK cell vs. NK-EV for immunotherapy.

	Pros	Cons
NK CELL	Origin of multiple and complex immune functions that includes NK-EVs. Able to incite stronger humoral response (antibody-mediated) via cross-talks with DC. Have explicit evidence on safety and references to efficacy. Have GMP-compliant isolation and expansion protocols.	Size and nature of cells limits permeability through the biological barriers. Poor cell viability and stability due to susceptibility from TME and TD-EVs. Limited modification or drug delivery potential. Possible graft rejection and transmission of infectious agents with the use allogeneic cells. No known effective cryopreservation methods despite years of research and development. Cell therapy remain an ethical issue. Low yield/purity and high cost of manufacturing.
NK-EV	Inherit the identity and properties of the parent cells. Nano-sized dimension improve homing and migratory function. Higher biostability, lower safety risks and excellent treatment efficacy. Unaltered by TME and/or TD-EVs. A viable drug delivery platform through genetic engineering or drug loading for an enhanced or combined therapy. Potential of storage with long shelf-life. Allow standardization of dose for targeted and precision medicine. Its classification as “biologics” is familiar to physicians and more ethically acceptable.	Requires new isolation from tissue and cell cultures to replenish stock. Lack of GMP-compliant protocols for product isolation, testing and preservation. Unknown clinical potential as a relatively novel product.

## Data Availability

Not applicable.

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
