# Peer review of "Natural Killer Cell-Derived Extracellular Vesicles as a Promising Immunotherapeutic Strategy for Cancer: A Systematic Review"

_ijms, 2023, doi:10.3390/ijms24044026_

Round 1

Reviewer 1 Report

Dear Authors,

There are few minor corrections.

line 79 its cross-linked

line 185 resemblance 

check line 300

line 364 lyophilized

In figure 4A and D, 7A please mention what does the numbers denote in legends and would be clear if cyan color is replaced or numbers are labelled in other color.

Author Response

Comment(s) from Reviewer 1:
Dear Authors,

There are few minor corrections.

Specific comments:
1. line 79 its cross-linked.

Answer: Thank you. The word has been corrected [Page 2: Line 79].

  1. line 185 resemblance.

Answer: Thank you. The word has been corrected [Page 7: Line 185].

  1. check line 300

Answer: Thank you. The sentence has been paraphrased and simplified [Page 12: Line 300-302].

  1. line 364 lyophilized.

Answer: Thank you. The word has been corrected [Page 14: Line 365].

  1. In figure 4A and D, 7A please mention what does the numbers denote in legends and would be clear if cyan color is replaced or numbers are labelled in other color

Answer:  Thank you. A sentence was added to end of each legend for Figure 4 [Line: 263-264] and Figure 7 [Line: 446 – 447] denoting the number in the pie charts. Furthermore, the cyan colour is replaced to ensure clear visual of number.

Reviewer 2 Report

The Manuscript by Alvin Man Lung Chan and co-authors is comprehansive and contains a lot of data on NK-derived EV. 

Strong sides of this work: 

data organized in series of tables (regarding in vivo and in vitro activity of NK-EV, also ways of isolation and characterization). 

What could be improved:

Side-by-side comparison of NK cells vs NK-EV as a tool to treat cancer. Pros and cons of NK cells and NK-EV.

More information on challenges with NK-EV production and stability/shelf life.

Author Response

Comment(s) from Reviewer 2:
comprehansive and contains a lot of data on NK-derived EV. 

Strong sides of this work: 

data organized in series of tables (regarding in vivo and in vitro activity of NK-EV, also ways of isolation and characterization). 

What could be improved:

Side-by-side comparison of NK cells vs NK-EV as a tool to treat cancer. Pros and cons of NK cells and NK-EV.

More information on challenges with NK-EV production and stability/shelf life.

Answer: Thank you for the suggestion. Table 4 [Page 18: Line 449 - 450] has been added to compare the pros and cons of NK cells and NK-EVs.

Reviewer 3 Report

This paper reviews an immunotherapeutic treatment strategy for tumors using immune cell derivatives.  The goal of this work is to further develop the potential for immunotherapy in treating a variety of tumors.  This is an important topic that hopefully will lead to prolongation of survival in patients with a variety of tumors.  The vesicles employed in this technique are an acellular product which appear to be resistant to factors in the tumor microenvironment that inhibit anti-tumor immune responses.  Some data regarding the potential efficacy of this treatment strategy with a variety of tumors in animal models is reviewed.  However, at this point it is likely that the clinical efficacy for this treatment strategy is speculative at best. 

Author Response

Comment(s) from Reviewer 3:

This paper reviews an immunotherapeutic treatment strategy for tumors using immune cell derivatives.  The goal of this work is to further develop the potential for immunotherapy in treating a variety of tumors.  This is an important topic that hopefully will lead to prolongation of survival in patients with a variety of tumors.  The vesicles employed in this technique are an acellular product which appear to be resistant to factors in the tumor microenvironment that inhibit anti-tumor immune responses.  Some data regarding the potential efficacy of this treatment strategy with a variety of tumors in animal models is reviewed.  However, at this point it is likely that the clinical efficacy for this treatment strategy is speculative at best.

Answer: Thank you for the comments.

Round 2

Reviewer 3 Report

This paper reviews an immunotherapeutic treatment strategy for tumors using immune cell derivatives.  The goal of this work is to further develop the potential for immunotherapy in treating a variety of tumors.  This is an important topic that hopefully will lead to prolongation of survival in patients with a variety of tumors.  The vesicles employed in this technique are an acellular product which appear to be resistant to factors in the tumor microenvironment that inhibit anti-tumor immune responses.  Some data regarding the potential efficacy of this treatment strategy with a variety of tumors in animal models is reviewed.  However, at this point it is likely that the clinical efficacy for this treatment strategy is speculative at best.